# The Smart Nervous System for Cracked Concrete Structures: Theory, Design, Research, and Field Proof of Monolithic DFOS-Based Sensors

**DOI:** 10.3390/s22228713

**Published:** 2022-11-11

**Authors:** Łukasz Bednarski, Rafał Sieńko, Tomasz Howiacki, Katarzyna Zuziak

**Affiliations:** 1Faculty of Mechanical Engineering and Robotics, AGH University of Science and Technology in Krakow, Mickiewicza 30, 30-059 Krakow, Poland; 2Faculty of Civil Engineering, Cracow University of Technology, Warszawska 24, 31-155 Krakow, Poland; 3SHM System Sp. z o.o., Sp. kom., Libertów, ul. Jana Pawła II 82A, 30-444 Kraków, Poland; 4Nerve-Sensors, Libertów, ul. Jana Pawła II 82A, 30-444 Kraków, Poland

**Keywords:** distributed fibre optic sensing DFOS, composite sensors, monolithic sensors, strains, cracks, widths, detection, reinforced concrete, laboratory, civil engineering

## Abstract

The article presents research on the performance of composite and monolithic sensors for distributed fibre optic sensing (DFOS). The introduction summarises the design of the sensors and the theoretical justification for such an approach. Lessons learned during monitoring cracked concrete are summarised to highlight what features of the DFOS tools are the most favourable from the crack analysis point of view. Later, the results from full-size laboratory concrete specimens working in a cracked state were presented and discussed in reference to conventional layered sensing cables. The research aimed to compare monolithic sensors and layered cables embedded in the same reinforced concrete elements, which is the main novelty. The performance of each DFOS nondestructive tool was investigated in the close vicinity of the cracks—both the new ones, opening within the tension zone, and the existing ones, closing within the compression zone. The qualitative (detection) and quantitative (widths estimation) crack analyses were performed and discussed. Finally, the examples of actual applications within concrete structures, including bridges, are presented with some examples of in situ results.

## 1. Introduction

### 1.1. Structural Safety

An acceptable level of safety is an obvious and essential requirement for structural design, construction, and maintenance. During each phase, relevant control procedures should always be specified and applied to ensure optimal decision-making [1,2]. These decisions are mainly about finding the balance between financial savings and keeping the failure risk at an acceptable level. However, this task is particularly important and, at the same time, challenging for:**ageing infrastructure** [3], which is many years old and in a technical condition problematic for correct assessment,**new engineering structures** with unusual geometry, construction technologies or material solutions for which there is a lack of common experiences.

Because of the complexity of civil engineering projects, including material and geometrical imperfections, uncertainties, time-depended effects, construction technologies and many others, structural safety cannot be ensured only by theoretical considerations, even taking into account the newest achievements in advanced numerical simulations. The knowledge must be developed based on the analysis of real structures’ behaviour, provided by in situ measurements. This task is currently being handled by structural health monitoring (SHM) systems [4,5] operating based on different sensing technologies selected according to a given project’s specific requirements. Usually, sensors for measurements of various physical quantities, like strains, displacements, inclinations, accelerations, forces, and others, are installed within the structure or construction site to monitor changes in these values over time during changing loading and thermal conditions. A well-designed system should provide reliable data for structural assessment, validation of simplified models, and decrease the risk of failure [6]—meaning an increase in safety.

It must be emphasised that failure consequences in civil engineering and geotechnical facilities include not only financial, social, and environmental losses but also loss of human health or life. This is the reason for the constant development of new solutions improving structures’ reliability and long-term high-quality performance.

### 1.2. Distributed Measurements of Concrete

Despite the advantages of conventional (discrete) structural health monitoring systems, a direct answer to the question about the state (e.g., the crack state) of the structure between the measurement points is still not possible. The solution to this problem is provided by distributed fibre optic sensing (DFOS), enabling the measurements with spatial resolution so high that they could be treated as geometrically continuous from an engineering point of view. Figure 1 shows the scheme comparing the capabilities of conventional spot strain gauges and distributed linear sensors installed within the example damaged concrete element. While performing distributed measurements, it is impossible to omit any extreme values, so one of the main goals of structural health monitoring—direct damage detection and its size estimation [7]—can finally be fulfilled.

The concept of smart structures with integrated systems based on distributed fibre optic sensors has been intensively developed for several years. Instead of a single number obtained by standard spot gauge, these sensors measure the profiles of strains [9], displacements [10,11,12], temperatures [13] or vibrations [14] over the entire structural length. DFOS-based systems are an analogue of the human nervous system [15,16], as the linear sensors can be compared to the individual nerves informing of dangers at any point and the data logger to the brain processing the data.

The above feature is especially useful regarding concrete because of its heterogeneity, local imperfections, changes in stiffness caused by the random presence of aggregate and dependence on many other factors like mixture components, time (rheology), curing process or load procedure. The most crucial aspect that should be underlined, though, is concrete discontinuities in the form of multiple cracks. They are a common and natural feature of reinforced and prestressed concrete structures [17]. Cracking already starts during early-age concrete hydration, when thermal-shrinkage strains are blocked by internal (reinforcement) and external (other elements, formworks, supports, boundary conditions) constraints [18,19]. Later, under operating conditions, mechanical loads, external temperature impacts and time-dependent structural degradation, cracks develop and change their widths. Knowing the actual crack state of concrete structures is one of the essential requirements for technical assessments of their condition and safety.

Comprehensive data gathered by distributed fibre optic sensors, replacing thousands of conventional spot-type sensors, provide a new quality in structural measurements and become the reason why DFOS nowadays is finding a growing acceptance in the laboratory, and above all, in field applications. It is worth underlining that it is possible and favourable to install the DFOS sensors inside the structural components [20] during construction, e.g., embedding them inside the concrete [21,22], which brings many benefits, including:The possibility of analysis of the structural behaviour from a real zero stress-strain state (usually, conventional gauges are installed within existing structures with the unknown initial level of stress and deformation).Integration inside the structure (concrete) provides a more accurate transfer of the measured physical quantity from the structure to the sensor (no additional mounting brackets or installation methods are needed).Natural protection of the embedded sensors against mechanical damages, aggressive external environmental or direct sunlight. The expected operation lifetime of such a system should be comparable with the operation lifetime of the structure itself.

### 1.3. DFOS Techniques (Data Loggers)

The general idea of distributed sensing described above can be utilised using different techniques reflected in the data loggers available on the market. Three main optical phenomena used for that purpose are Rayleigh, Brillouin and Raman scatterings, named so to honour their discoverers (Figure 2). It is worth noting that both Rayleigh and Raman were awarded the Nobel Prize in the early twentieth century for their achievements in physics.

Each of these approaches is characterised by its own advantages and limitation related to measurement parameters. The final choice should always be preceded by a detailed analysis of the requirements of the project in question. However, general insights can be presented as follows:**Rayleigh scattering** [23] is used to measure the strains in optical fibre, which are caused both by mechanical and thermal loads. It provides the best spatial resolution, starting from the mm range [24] (1000 measurement gauges per 1 m of the linear sensor), which is particularly useful when analysing localised events like cracks. Moreover, dynamic readings with a frequency of up to 250 Hz are also possible [25]. However, the main limitation of that approach lies in a measurement distance range limited to 100 m (while keeping the high spatial resolution).**Brillouin scattering** [26,27] is used to measure the strains in optical fibre, which are caused both by mechanical and thermal loads. It provides multiple lower spatial resolution (from 20 to 100 cm) compared to Rayleigh scattering. On the other hand, it allows the measurements to be performed over very long distances (e.g., 25 km or more). This makes it suitable for monitoring linear structures like tunnels, roads, embankments, dams or mining and landslide areas.**Raman scattering** [28,29] is used to measure the temperature profile in optical fibre so that it could be used as one of the compensation solutions for Rayleigh and Brillouin measurements. Standard spatial resolution starts from 50 cm, while distance ranges from 25 km and more. This approach can be successfully used as a thermo-detection method [30] to localise fires or leakages.

It must be underlined that thanks to the dynamic development of commercially available optical solutions, their parameters are constantly improved. It is also possible to apply a few DFOS techniques to read the same sensors and thus, obtain extended benefits like automatic thermal compensation [31] or simultaneous measurements of different physical quantities.

### 1.4. DFOS Sensing Tools

The efficiency of the entire DFOS-based system depends not only on the optical data loggers’ parameters but also on the quality and performance of the measuring tools. Optical devices are usually designed by physicists, optical and electronics engineers, and IT specialists. However, the design of the tools being installed within the concrete, very often in challenging construction conditions, requires the knowledge of civil engineers, materials scientists and mechanics.

There are three main groups of DFOS sensing tools [6], including optical fibres in their primary coating, layered sensing cables and monolithic, composite strain sensors (Figure 3). The first group is mainly used in laboratory conditions because of the dimensions of standard optical fibre (outer diameter of 250 µm—Figure 3a), which makes it susceptible to break or be disturbed by any local and transverse pressures. That is why installation conditions must be strictly controlled, which is impossible on construction sites or geotechnical fields.

A few aspects must be considered regarding the measurements of cracked concrete. The basic thing is the type of the primary coating, which influence the maximum strain range, strain transfer mechanism or the resistance of the fibre to an alkaline concrete environment. Two commonly used primary coating types are acrylate (softer) and polyimide (stiffer). The final choice should be made considering the following relationships:the softer the coating, the higher the maximum strain range [6,20] (during crack detection, very high and local strain values are expected, so the acrylate coating minimises the risk of the fibre’s breakage);the stiffer the coating, the better the strain transfer mechanism and the shorter the length of strain mobilisation [32,33] (it means that acrylate coatings are not preferable for short measurement sections);alkaline concrete environment degrades the polyimide coating even after 14 days, while no influence is observed on the acrylate one [34] (that is why polyimide coatings are not advised for long-term measurements of concrete structures).

To enable safe handling of the measurement fibre on site, it must be somehow protected. There are two basic concepts utilised in practice. The earliest approach involves techniques known from telecom applications, where a set of intermediate layers are arranged around the fibre for its mechanical protection. In the worldwide literature, these solutions are known as layered sensing cables [35,36,37]. The layers are usually made of plastics, but steel strengthening inserts are sometimes applied (Figure 3b) to improve the strength parameters of the entire cable. However, it is important to note that the yielding point of steel is approximately equal to 2000 µε, while the maximum range of the fibre itself could exceed even 50,000 µε [6]. Moreover, crack-induced strain values are usually multiple times higher that yielding point of steel [6,8].

The presence of intermediate layers implicates the risk of mutual slippage between them and thus, influences the correct strain transfer from the concrete to the measuring fibre inside the cable. The problem of slippage is especially noticeable during high strain values and high strain gradients close to the cracks. This phenomenon was observed and described in many works [6,38,39,40,41], confirming that the presence of a coating or layer results in differences between the structural strains and those sensed by the cable.

Another uncertainty is caused by thermal changes, which, in the case of different and unknown thermal expansion coefficients of subsequent layers, may produce unknown strains in the fibre. Thus, the mechanical and physical properties of the cable’s components and their adhesion parameters must be carefully studied before application within the cracked concrete structures to enable the correct data interpretation. Moreover, the name ‘cable’ corresponds to the feature of these tools, i.e., their usability only when subjected to tension loads. If compression strains are expected, the pre-tensioning process should be performed during installation [42], which is difficult, and sometimes even impossible, in the case of site or geotechnical conditions.

Finally, the type of outer surface of the cable should be considered to provide its best possible bonding between the surrounding concrete. Figure 3e shows the example surface with tiny, perforated grooves. However, most conventional cables’ outer surfaces are entirely smooth, which increases the unfavourable effect of internal slippage.

Another concept for fibre protection is proposed by monolithic sensors, which are produced as composite bars in the pultrusion process [43]. The glass measuring optical fibre is fully integrated within the monolithic core without any intermediate layers (Figure 3c). Thus, any potential disturbances caused by the internal slippage within the DFOS tool are excluded. External unidirectional (Figure 3e) or bi-directional braid should provide the best bonding properties with surrounding concrete, similarly to reinforcing bars (not only by adhesion but also by mechanical clampings).

Although the general idea and operation rule of monolithic sensors remain the same, their properties could be adjustable depending on the specific requirements of the project in question. Both geometrical (diameter, cross-section shape) and mechanical (strength, elastic modulus, limit strain) parameters can be considered. The design of a DFOS-based system involving monolithic sensors for crack detection in concrete is presented hereafter and discussed based on actual data from laboratory and field tests.

The research shows the performance of monolithic sensors when monitoring cracked concrete structures in comparison with selected layered cables. All of these tools are commercially available on the market and widely used, so the presented results would have important meaning for practical applications in the future.

## 2. Design of DFOS-Based System for Cracked Concrete

### 2.1. Main Objectives

The main goal of distributed fibre optic sensing for cracked concrete is to enable the advanced analysis of cracks. However, it should be clarified what this analysis means in practice. Two issues should be considered:**qualitative analysis**: cracks’ detection and location (Figure 4a);a significant limitation of spot techniques for crack measurements is that they can be used only when the presence and the position of the crack are known. Knowledge about the location of the crack is not necessary during distributed sensing because its detection is one of the system’s objectives. The possibility of crack detection was checked and confirmed in much research, e.g., [44,45,46]. However, this does not mean that the effectiveness of each DFOS system is unconditional and absolute.**cracks’ width estimation** (Figure 4b);knowledge about the crack’s presence and location is necessary but insufficient for assessing structural safety. The DFOS system should provide information about the actual width of the crack (mm), changing over time, which could be compared to the thresholds defined in relevant standards to answer the question about the crack’s significance for load capacity or durability. The measured strain profile in the close vicinity of the crack should be converted into the crack width (in the literature called also crack opening displacement COD) with reasonable accuracy, useful from a practical point of view (not worse than 0.05 mm). A few procedures are presented in the literature [47,48,49,50], taking into account applied spatial resolution and the assumed physical model of the entire system (also internal design of the DFOS tool).

### 2.2. Selection of DFOS Technique

Distributed sensing through a fibre optic sensor is supported by several optical phenomena already outlined in Section 1.3. The cracks formed within the concrete structures usually have widths much lower than 1 mm, so they should be treated as a local event. This is the basis for selecting the DFOS technique with appropriate high spatial resolution. Despite the crack widths or their changes reaching only the tenths of a millimetre, the spatial resolution of mm or cm order is sufficient due to the deboning effect between the DFOS tool and the concrete. This effect is similar to that observed during the interaction of the concrete and reinforcing bar within the crack (Figure 5a), where debonding length occurs over a few to several centimetres, influencing the crack-induced strain profiles. A simplified graphical interpretation of high spatial resolution defined for the sensor installed in concrete along the reinforcing bars is presented in Figure 5b. The bases and spacings of virtual gauges, which are connected in series in a chain, allows for the identification and analysis of localised events.

Today, the above requirements on high spatial resolution are usually met by Rayleigh scattering. For standard Brillouin-based systems, spatial resolutions range from 200 to 1000 mm, which is insufficient to solve the stated measurement problem (qualitative and quantitative crack analysis). However, it should be clearly emphasised that thanks to the dynamic development of DFOS techniques, the performance of optical data loggers (including spatial resolution) is constantly being improved. Other parameters should be counted when selecting the appropriate one, like accuracy, strain resolution, precision, or repeatability.

### 2.3. Design of DFOS Tools

DFOS tools are a critical component of the entire system. They are fully integrated with the monitored concrete structure (by mounting within the surface or embedding inside the structural member) to provide reliable measurements during the predicted period. Preferably, this time is equal to the operational lifetime of the structure itself. In most projects, it is difficult, and sometimes even impossible, to replace the DFOS tools, while the reflectometer can be easily replaced in case of failure.

The high-quality and long-term performance of DFOS tools dedicated to civil engineering and geotechnical applications must be ensured by meeting the set of the following requirements:**High accuracy** ensured by unambiguous strain transfer from the structure to the measuring fibre inside the DFOS tool. This feature is characteristic of a monolithic cross-section of the sensor without any intermediate layers, which disturb measurements by extensive slippage. The lack of slippage within the sensor itself allows for reducing uncertainties and simplifying mathematical models used for strain transfer analysis and crack width calculation;**High strain range** allowing for undisturbed readings of crack-induced deformations without fear of damaging the sensor’s components. Fast-yielding materials like steel or plastic tend to remember the localised historical strains rather than reflect the actual deformation state of the structure. This is especially dangerous during long-term monitoring while cyclic loads are expected;**Rough outer surface** of the sensor must provide the best possible bonding with the surrounding concrete, not only through the adhesion but also mechanical clamping (similar to the reinforcing bars). For instance, it could be done by ribs, braids or perforated grooves [52];**Resistance to harsh conditions**, which are expected during the construction and operation of civil engineering structures. The design of the sensor must provide appropriate protection against mechanical damages, local transverse forces (e.g., pressure of aggregate grains or mounting elements), alkaline concrete environment and other aggressive factors;**High durability** provided by appropriate material. As the sensors are usually fully integrated within the concrete, the expected operation lifetime should be equal to the lifetime of the structure itself;DFOS tools **cannot require pretension**, as this process in construction or geotechnical conditions is challenging or often impossible. Selected stiffness of the sensor’s core must ensure correct positioning without extensive waving, as deviations from the designed position will result in additional errors during data interpretation. Reliable readings must be possible both in the tension and compression zone;**Prove the sensors’ high performance** in at least tens of engineering projects.

Debonding effect between the outer surface of the sensor and the surrounding concrete is natural and cannot be entirely eliminated. Perfect bonding means the full compliance of concrete and sensors strains, but in case of concrete discontinuity (crack), it would result in the sensor’s breakage. Thus, debonding over the section called “effective length” appears, reducing the infinite theoretical strain to the level which will not cause the sensor’s breakage. This is how the reinforcement bars work in the close vicinity of the crack (Figure 5a). Designing monolithic sensors for crack detection and analysis is thus about finding the balance between bonding quality, the core’s elasticity and its maximum strain.

Example parameters of two different monolithic strain sensors are summarised in Table 1 for comparison.

Monolithic sensors can replace the composite reinforcing bars and be included in strength calculations of the structure itself. Figure 6a shows the footbridge example [15], where stiff monolithic sensors (called “EpsilonRebars”, Ø5 mm, E = 50 GPa) were used to reinforce the hybrid deck over the entire length of 80 m.

On the other hand, to increase the sensitivity in crack detection and to avoid strengthening the structural elements, it is favourable to reduce the axial stiffness as much as possible [6]. Figure 6b shows the example of a laboratory reinforced concrete slab, where the applied sensors (called “EpsilonSensors”, Ø3 mm, E = 3 GPa) provided an almost 50 times reduction in stiffness, being invisible to the structure itself.

### 2.4. Installation Methods

After selecting the DFOS technique and type of monolithic sensor, the next important issue influencing the system’s reliability and efficiency is the installation method. It depends on the type of structure (small-size laboratory specimens or full-scale engineering facilities) and the mounting time (existing, ageing infrastructure or new one).

The common approach for existing structures is to glue the sensors to the prepared (grinded, cleaned and degreased) concrete surface [6,52]. The monolithic sensors can be produced as flat bars (tapes) to facilitate that process and increase the bonding surface. However, thinking about long-term measurements, the more favourable solution is to install the sensors in pre-made near-to-surface grooves [52] filled with two-component epoxy or mortar injection (Figure 7a). This approach is preferable due to:the maximum bonding surface between the sensor and the surrounding concrete (from three sides instead of one in case of surface installation);natural protection against mechanical damages;significant reduction of thermal influence (direct sunlight) on strain results;the beast aesthetics without mounting elements visible on the surface.

The adhesive parameters, similar to the sensors themselves, should be selected carefully to ensure the best strain transfer mechanism. Too stiff adhesive can crack itself, while too soft can underestimate the actual structural strains.

Sensors can be installed not only within the concrete surface but also on the reinforcing bars along their longitudinal rib (Figure 7b). Based on that approach, crack detection will still be possible, but the main goal will be rather to assess crack-induced stress in the reinforcement instead of precise width estimation (because of the very high stiffness of reinforcing bars, the effective length of debonding will be higher in comparison to the effective length of the flexible sensor itself).

The most convenient way of installation is applied in the new, reinforced structures. The sensors cut at the required length are usually delivered on site in coils, and the only task is to unroll them and stabilise them in designed positions by tying to the existing reinforcement using cable ties (Figure 7c). The significant advantage of the approach is that the measurements can be started at real zero stress-strain-crack state and being performed through all the constructions stages [53]: from early-age concrete, thermal-shrinkage strains, and corresponding micro-cracks, through prestressing and construction stages to the load tests and final operation. Structural assessment can be done more effectively based on such data because they refer to the absolute values of measured parameters (strains or crack widths) instead of their changes (increments).

### 2.5. Thermal Compensation

The measured strain profiles using Rayleigh or Brillouin scattering are affected both by mechanical loads and thermal changes. In most general terms, the following equation can be presented:(1)Δε(x)=f(ΔL(x),Δ T(x))
where Δ*ε*(*x*) is the strain measured over the distance *x*, Δ*L*(*x*) is the mechanical strain change and Δ*T*(*x*) is the temperature change.

Providing that temperature changes between the subsequent readings are constant, measured strains are directly equal to the mechanical strains (caused by external forces), i.e., generating the stress inside the structural component. This situation is possible only in laboratory conditions or during short-term field measurements, especially overnight or within underground installations.

However, during long-term monitoring, knowledge of thermal changes is unconditionally necessary to enable the correct engineering interpretation of structural behaviour. Appropriate correction can be applied based on the technical specifications of monolithic sensors and the guidelines of their producers. Usually, in the standard operating range (from −20 °C to +60 °C), it is enough to use a single (linear) coefficient, including the sensor core’s thermal expansion and the refractive index’s temperature-dependence of the optical fibre itself. This task became more complicated and uncertain with coated fibres [31] and layered cables, as the individual coatings have different (usually unknown) thermal expansion coefficients. With temperature changes, they interact with each other causing additional mechanical strains in the measuring fibre.

Nevertheless, the profiles of temperature changes between the subsequent readings over the entire monitored length are the basis for further compensation procedures. Today, there are four possible approaches to obtaining them:Using the strain DFOS sensors with a Raman-based optical datalogger, which is insensitive to mechanical loads and thus, allows only for temperature measurements; DTS (distributed temperature sensing). This solution is primarily dedicated to long distances (km order).Using the strain sensors with both Rayleigh-based and Brillouin-based optical datalogger. These two techniques are depended on mechanical strains and temperatures to varying degrees. Knowing the individual coefficients for each technique, it is possible to solve a system of equations in which the unknowns are mechanical strains and temperatures [31]. It is worth noticing that there are already hybrid data loggers available on the market that use different optical phenomena in their design.Using special DFOS temperature sensors and one from DFOS techniques for strain measurements (Rayleigh or Brillouin). The idea behind this solution is to isolate the measuring optical fibre from mechanical strains, for instance, by placing it inside the tube. The fibre is then, apparently, subjected only to temperature changes. However, the free fibre does not exist because of the friction between the tube and the fibre. It can cause disturbances in the measured temperature profiles, especially considering longer distances. In addition, high mechanical strains expected while monitoring the cracks in concrete can consume excess fibre inside the tube.Using conventional spot temperature gauges and approximating the temperature field between the measurement points. That approach is justified when no high gradients over length are expected (like in underground installations or other horizontal sections with similar conditions over the entire length).

### 2.6. System’s Design—Summary

Multiple aspects must be considered when designing the DFOS system for crack monitoring in concrete structures. After selecting the DFOS technique, appropriate sensors, installation methods and thermal compensation approach, the last thing is to elaborate algorithms and procedures for data post-processing, allowing the system’s main objectives to be achieved. First, raw strain data should be validated, and eventual distortions should be removed. For the inexperienced user, cross-correlation failed points can be misinterpreted as cracks [7]. Then, thermal compensation should be applied to distinguish between the strains causing the mechanical stress in the element (*ε_σ_*) and those related to the change in length (*ε_L_*)—actual shortenings or elongations consist of the stress-free part (unconstrained thermal strains) and part generating the stress. For correct crack width estimation, strains related to the real change in length should be considered. For example, for ultimate limit state analysis, only strains generating the stress are essential.

As described in Section 2.1, there are a few methods for crack width estimation and choosing the appropriate one should be preceded by the studies on uncertainties and possibilities of applied software (in case of automatic analysis). It is recommended to define thresholds in reference to the relevant standard and the decision-making procedure after possibly exceeding these values. One of the important steps is the intuitive representation of the data, both in length and time domain. The convenient way is to present the strains in relation to the geometry of the structure in question, which makes physical interpretation easier.

The proposed step-by-step procedure of designing a DFOS system dedicated to crack monitoring in concrete structures, with general descriptions, is summarised in Table 2.

## 3. Laboratory Tests

### 3.1. The Concept and Preparation of the Specimens

The laboratory tests described hereafter were one of the first in the world, allowing for direct comparison between different types of DFOS tools embedded in the same reinforced concrete beams. A total of six beams with rectangle cross-section 250 × 350 mm were made, with tools installed in the lower (tension) and upper (compression) zones. The spatial visualisation of the single specimen is presented in Figure 8a. Two types of layered cables and two types of monolithic sensors commercially available on the market were selected (Figure 8b), which makes the results particularly important in terms of practical applications:C1: sensing cable with three plastic layers (Ø2.8 mm, E = unknown).M1: monolithic, reinforcing sensor (Ø5.0 mm, E = 50 GPa)M2: monolithic, flexible sensor (Ø3.0 mm, E = 3 GPa)C2: sensing cable with two plastic layers and steel insert (Ø3.2 mm, E = unknown).

To avoid damage or changing the position of the tools during concreting, self-compacting concrete was applied (Figure 8c). There were three beams with 10 mm main reinforcing bars (named “S”) and three with 20 mm (named “L”). Thanks to that, different crack patterns (including the cracks’ number, widths and spacing) were obtained.

Another implemented approach was to use a concrete mixture with a large amount of Portland cement and to leave the top surface of the beams after concreting without any curing (Figure 9a). The idea behind that procedure was intended to intensify the shrinkage process and thus, the formation of cracks within the upper part of the beams, what was finally succeeded (Figure 9b). Thanks to that, during mechanical tests, the effectiveness of DFOS tools was analysed in relation to the new cracks being opened in the tension zone (Figure 9c), simultaneously with the existing cracks being closed in the compression zone.

### 3.2. Course of the Study and Measurements

The research was performed in the Building Materials and Structures Research Laboratory at Cracow University of Technology, Poland. Beams were loaded in four-point bending tests using the universal machine with the max. load cell capacity of 1000 kN, which is permanent equipment of the laboratory (Figure 10a). The machine’s accuracy is ±0.5% of the reading down to 1/500 of the full scale, while the maximum force applied during research was equal to 210 kN. The static scheme of the simply supported beam has been provided by rolled supports, allowing for both rotations and horizontal movements.

The DFOS readings were done statically step-by-step (while increasing force) using a Rayleigh-based datalogger with an optical switch connected to all embedded tools inside the beam (Figure 10b). During selected load steps, the reference readings were performed by portable optical microscope with a resolution of 0.02 mm on the side surface of the beam (Figure 10c). An in-built scale above the focusing ring allowed for convenient readings. The results were systematically written on the beams using a marker. This method was applied for the crack widths up to 0.4 mm, while the plate crack-meters were used for the wider ones.

The test course has been established separately for S- and L-type beams due to their different structural response resulting from different reinforcement (4 times smaller cross-sectional area of main bars for beams type S). Due to the many beams, load steps and multiple cracks formed during research, in the following part of the article, the entire procedure and results will be presented and discussed for the example beam S3 for the sake of transparency. In the beginning, the experiment was steered (controlled) by the displacement of a hydraulic cylinder, which was stopped at selected force values. Figure 11 shows the measurement schedule graphically with designed load cycles. Blue dots represent the load steps when DFOS measurements were taken, while green arrows indicate the reference readings of crack widths. In S-type beams, after reaching the force of 55 kN (27th load step), it was impossible to keep it constant when the piston stopped. Thus, it was decided to move the piston 2 mm down at each load step, no matter the force value. It was because of the significant and uncontrolled crack development and reduction in stiffness.

All the beams were loaded up to the structural failure, which was achieved by losing the bending capacity in S-type beams and shearing capacity in L-type beams.

### 3.3. Compression—Example Results

A detailed comparative analysis of all DFOS tools in crack detection and width estimation will be the subject of another article [54]. This paper chose two fundamentally different tools for the abbreviated presentation: a monolithic sensor with reduced axial stiffness (M2) and a sensing cable (C2) with two plastic layers and a steel insert. That approach aims to underline the differences resulting from the tool’s internal design. In the following part of the article, the results for the monolithic sensor will be represented by the red scale lines, and for the layered cable, by the blue scale lines.

The graphs presented in Figure 12 show the strain profiles obtained over the entire length of the beam (4 m) within the compression zone, where existing shrinkage cracks were closing under mechanical load. Presented data refer to the first twenty-seven measurement steps (up to the force of 55 kN), in which the beam behaviour was stable and the results, therefore, comparable. The zero reading was assumed just before the loading (the dead weight is not included). Negative strain values correspond to the increase in compression, while the positive ones to the tension. The closing cracks induced all the local negative strain peaks. Five of them were chosen for further analysis and marked on the plots. The maximum registered strain was approximately equal to only 550 µε because the width changes of the existing cracks in the compression zone, in contrast to the opening cracks in tension, are limited by the initial crack width.

Despite the minor differences in the shape of strain profiles, which could be caused by the random course of the cracks and their varying size over the beam’s width, it should be stated that analysed tools could detect all the existing cracks correctly. The locations of strain peaks were in line with the external observations. Moreover, the crack width changes were calculated by strain integration over the effective length, and the maximum difference between the tools during subsequent load steps did not exceed the value of 0.01 mm, while the mean difference was a few times smaller. Such a value, treated as an error in width estimation, allows for correct structural safety assessment and can be usually neglected from an engineering point of view.

### 3.4. Tension—Example Results

Although the results provided by all the analysed DFOS tools in the compression zone were satisfying, it should be underlined that the obtained strain range was minimal in comparison to the declared maximum strains (±40,000 µε for sensor M2 and ±10,000 µε for cable C2). Moreover, while analysing the behaviour of reinforced concrete structures, the tension zone is of the main interest, including cracks increasing their widths.

Figure 13 shows the strain profiles in the tension zone over the entire structural length during subsequent load steps, corresponding to those presented in Figure 12 for the compression zone. The scale of the horizontal and vertical axes is the same to facilitate visual comparison. For the load step no. 27 and the corresponding force of 55 kN, the maximum strain value within the crack-induced area registered by the monolithic sensor was equal to 16,780 µε, while by the layered cable, it was almost four times smaller—4569 µε. Given that both tools are theoretically dedicated to the same purpose and that the above maximum values are still well below the declared limits, such a difference in strain values is not acceptable.

The above observation about a significant decrease in the crack-induced strain peaks in the tension zone is valid for all the beams and all the cracks. Figure 14 presents the maximum strain values changes in subsequent load steps within the five selected cracks, which locations are marked in Figure 13.

Because the strain data gained through distributed sensing are expressed in two domains: structural length and time (load step), they could be conveniently presented in the form of spatial visualisations. The example plots corresponding to data in Figure 13 for monolithic sensor M2 and layered cable C2 are shown in Figure 15a and Figure 15b, respectively.

The locations of all the cracks were identified unequivocally by a monolithic sensor with an accuracy equal to applied spatial resolution (10 mm), which was confirmed by reference readings using a metric tape on the side surface of the beam and photo-camera. On the other hand, the indications from the layered cable are not conclusive, especially given the negligible increments during cyclic loading. In real projects, very often, the zero reading is taken after the formation of the cracks and width changes are usually much smaller than in laboratory conditions (under the applied force). Thus, the risk of data misinterpretation is too high. Figure 16 summarises on the same plot the strain profiles measured by monolithic sensor (red line) and layered cable (blue line) in load step no. 26 (50 kN), in reference to the documented cracks’ locations on a side surface.

The next step of the analysis was focused on the crack width estimation, which was done by strain integration over the effective length. This length was assumed to be equal to half of the mean spacing between the multiple cracks (the conservative assumption about overlapping the effective lengths). The local crack-induced strain peaks indicated the centre of this length. The crack widths calculated for the example load step no. 26 are summarised in Table 3, showing that the cable’s results underestimate the sensor’s results by almost 0.15 mm (≈30%). In addition, the results from the external microscope were provided to give the general overlook on the actual crack widths. However, these values cannot be compared directly, as the readings were taken in different place (on the side surface).

### 3.5. Findings

Considering the above specific example, but also global statistics presented in [7], the most sensitive and reliable tool for crack detection and analysis is a monolithic sensor with reduced axial stiffness. Unambiguous measurements are further supported by theory: the internal design of the cross-section includes the full integration of measurement fibre with composite core during its pultrusion, eliminating the slippage phenomena. Moreover, minimal axial stiffness means that the sensor is no longer the reinforcement for the concrete and does not influence its structural response.

Another advantage is the highest strain range. The maximum value of tensile strain registered in the last load step, just before the structural failure, was equal to 2.86% (28,575 µε), while the strain range declared by the producer is ±40,000 µε. The corresponding crack width exceeded 1.3 mm at that time. The ability to measure extensive cracks without fear of damaging the sensor is an essential aspect while monitoring cracked reinforced concrete structures.

Although the layered cable with steel insert inside provided reasonable results in the compression zone in the very limited strain range (<550 µε), its performance was unacceptable within the tension zone. The registered crack-induced strain peaks were almost four times lower compared to the monolithic sensor, and the shape of strain profiles did not allow for certain crack detection. The maximum declared value of correct strain measurements (equal to 1%) is not feasible. It is worth noticing that the steel layer inside the cable yields itself at the strain level of approx. 0.2%. The main reason for unreliable readings, though, is disturbed strain transfer from the concrete to the measuring fibre caused by the internal slippage between the intermediate layers in the cable.

Considering the above findings, the layered tools are not recommended for high-spatial-resolution DFOS measurements of reinforced concrete structures when crack detection and analysis are of the main interest.

## 4. Field Proofs

Field proofs are the last and most important step in verifying the effectiveness of DFOS-based monolithic sensors. This section describes three examples of actual concrete structures equipped with such sensors, giving the example results related to crack analysis. The first is a new bridge with sensors embedded inside the concrete slab. The second refers to the existing 10-years-old bridge, where the near-to-surface installation method was applied. Finally, the very old concrete sewer collector is described as an example of an ageing infrastructure requiring the appropriate control and maintenance.

### 4.1. Railway Bridge near Frankfurt, Germany

Three monolithic sensors were used to monitor the performance of the concrete slab in a railway bridge north of Frankfurt am Main, Germany. Since the decision to create the system was made before construction was completed, it was possible to embed the sensors inside the slab. The installation consisted of stabilising the sensors in their designed positions by tying them to the existing reinforcement along the main bars (Figure 17).

Sensors with appropriate lengths were delivered on site in coils, and the ease of installation allowed it to be completed within half a day. As the sensors were installed before concreting (Figure 18a), the structural performance analysis was possible from an actual zero state, i.e., zero readings were taken before the first cracks were formed. The blue lines on the strain plot in Figure 18b correspond to the measurements performed a few hours after concreting. There are still no characteristic crack-induced strain peaks. The data are presented over the 10 m long section.

However, the measurement performed the next day, 20 h after concreting is completed, clearly indicates the presence of the first ten cracks (red line on the plot). Their spacings and locations depend on the geometry of the structure and boundary conditions. The estimated crack widths did not exceed the value of 0.06 mm.

Cracking in concrete structures is a normal process taking place from the first days, and it usually cannot be avoided. The structural engineer’s main task is to monitor and control the crack positions and widths over time. Distributed fibre optic sensing allows all the cracks to be detected and analysed, even inside the structural elements, where visual inspection is impossible. However, a precondition for the system efficiency is the use of high-definition monolithic sensors, high-spatial-resolution DFOS technique and appropriate data processing algorithms.

The DFOS-based system in the Frankfurt bridge was used to detect the first cracks formed during early-age concrete behaviour (thermal shrinkage strains) but also to monitor structural response according to the planned schedule, including long-term static measurements as well as dynamic readings during the train passages.

### 4.2. Largest Concrete Cable-Stayed Bridge in Poland

Although embedding sensors inside the components of new structures is most beneficial from the quality and quantity of information obtained, it is also necessary to be aware of the existing, ageing infrastructure [55]. The following example is related to one of the largest concrete bridges in Poland—Rędziński Bridge [56,57] (Figure 19a). It was opened to traffic on August 31, 2011, and is the most significant bridge along the Wrocław motorway ring road. It is a four-span cable-stayed bridge over the Odra River with spans of 50 m + 2 × 256 m + 50 m long. The two separated concrete box girders are suspended to a single H-shaped pylon with a height of 122 m, which makes it the tallest in Poland. Its shape allows connecting 160 stays arranged in four planes.

Due to the limited width of the route below the lower crossbeam, the pylon’s legs are inclined to minimise the size of the foundation (massive concrete slab with a base of 67.4 × 28.0 m and thickness from 2.5 to 6.5 m, placed on 160 reinforced concrete bored piles with a diameter of 1.5 m and 18 m in length). The pylon’s legs have a rectangular cross-section with variable dimensions: 6.0 × 7.0 m at the foundation level, 4.0 × 4.0 m near the upper crossbeam and 4.0 × 6.0 at the top.

Although the decks are suspended by the set of cables, they are also (to a lesser extent) supported by the prestressed lower crossbeam of the pylon. However, the main task of this prestressed element, with the midspan rectangle cross-section of 2.5 m (height) and 4.0 m (width), is to connect two inclined legs of the pylon, which, due to their geometry, tend to move outwards (Figure 19b).

After ten years of the bridge’s operation, multiple cracks (with widths up to 0.1 mm) were observed over the length of this crossbeam, which is a natural consequence of its static scheme and loading. However, due to the unfavourable environment above the river, controlling cracks widths changes over time (according to the standard’s requirements) is very important.

The Rędziński Bridge was equipped with DFOS monolithic sensors in 2020, joining the honourable group of several bridges in Poland with this innovative monitoring solution. Four sensors were glued inside the pre-made near-to-surface grooves (Figure 20a) located in each corner of the lower crossbeam of the pylon (Figure 20b) over its entire length (Figure 20c). This installation approach is the best in terms of bonding properties and mechanical protection, but also in terms of reducing the direct impact of the sunlight and ensuring the appropriate aesthetics in the facility being operated.

The cyclic measurement sessions are now performed according to a planned schedule, with reference taken shortly after the installation. In a single reading, the strain data from 7400 locations are recorded, providing direct crack detection and estimation of their width changes (both while closing and opening).

Next to the long-term analysis, the short-term readings (at an interval of a few seconds) were also done under the random traffic loads. The width changes identified then did not exceed thousandths of a millimetre. Example results from the lower and upper sensor are presented in Figure 21. Strain profiles with crack width analysis are shown on a few metre-long sections, selected in reference to the structure’s geometry, to keep appropriate clarity.

Despite such small values of crack widths changes, the system was able for a reliable diagnosis thanks to the appropriate strain transfer from the concrete to the optical fibre fully integrated inside the monolithic core of the high-sensitivity sensors.

Four spot temperature gauges were installed within the crossbeam to compensate for the long-term thermal influences. It is also worth noticing that the applied strain sensors can be used simultaneously with other dataloggers (based on various optical phenomena) to measure different physical quantities like distributed temperatures or vibrations. Finally, thanks to the known arrangement of the sensors within the beam’s cross-section, it is possible to calculate horizontal and vertical displacements (deflections, shape changes), as well as axial shortenings or elongations.

In 2022, the force adjustment within eight shortest stays was performed, relieving the pylon’s lower crossbeam and bending it up slightly. Distributed fibre optic monolithic sensors were used for the advanced and simultaneous analysis of the beam’s strains, cracks and displacements.

### 4.3. Renovated Sewage Collector in Poland

The last example refers to the Burakowski sewage collector, an essential element of the sewage system in Warsaw, Poland [58]. It was built in the 1960s using the mining method in a concrete casing, with segments of lengths from 2 to 3 m and an internal diameter of approximately 3 m, buried under the ground surface at a depth of 4.5 to 7.5 m. Its designed capacity is about 12 m^3^/s, while the recorded capacity may be higher during heavy rainfalls.

The unfavourable external conditions in the close vicinity of the collector include the edge of the high bank of the Vistula River, as well as subway and tram lines and stations. In addition, several tall and deep-founded buildings have been built in this area over the last 15 years. In 2015, the construction of a new “Burakowski Bis” collector was completed passing along the existing one, which resulted in environmental changes (including ground-water conditions) compared to the times when it was constructed.

After analysing the collector’s technical condition in 2019, a decision was made to retrofit a 4.8 km long section using non-circular 3 m-long GRP module panels and relining technology. The space between the panels placed inside and the existing inner walls of the concrete collector was filled with a cement injection. Thus, the possibility of a visual inspection of the collector was lost.

This was one of the reasons for equipping the old concrete part with a DFOS-based system. Three monolithic sensors were installed longitudinal over the length of the 146 m-long section between the maintenance holes. One top sensor and two side sensors (Figure 22a,b) in the known configuration were mounted in near-to-surface grooves filled with mortar injection (Figure 22c). The approach, like this applied in the above bridge example, aimed for simultaneous analysis of strains, cracks and vertical displacements during the retrofitting process, but also after putting the collector to the normal operation again.

What is more, one monolithic strain sensor was installed along the circumference of the collector’s cross-section (Figure 23a), creating four loops located in the most safety-critical area (Figure 23b), where a few cracks were documented through visual inspection before the installation of the sensors (Figure 23c).

Strain measurements were taken during the retrofitting and strengthening works, including the following stages:before GRP modules were provided inside the collector (zero reading),after the GRP modules were placed (before grout injection),after grout (mortar) injection,and finally, after the collector was put back into service and filled with sewage.

DFOS readings allowed observing the collector’s structural response under gradual load changes.

The dead weight of the new panels and mortar injection, as well as the dead weight of the launched sewage flow, caused the original structure of the concrete collector casing was reflected in DFOS readings. Increasing mechanical load during the renovation work revealed cracks caused by technological breaks, and changes in their widths were carefully analysed during this process. Figure 24 shows strain profiles registered by the top sensor over a 10 m-long section in the subsequent load steps. Crack-induced strain peaks are negative, which means they correspond to the compression (decreasing the crack width). The maximum changes in crack widths during retrofitting did not exceed the value of 0.02 mm. The crack spacing results from the original construction technology of the collector, where concrete casing sections had the length from 2.2 to 2.9 m.

The application of the DFOS system based on monolithic strain sensors allowed for detection of all discontinuities in the collector structure, most of which were undetectable by the naked eye during visual inspection. The system is now used to control the crack widths over time during periodical measurement sessions, supporting the expert in assessing the technical condition and structural safety of the collector.

## 5. Conclusions

There has been a noticeable increase in distributed fibre optic sensing deployment in recent years, not only in laboratory research but, above all, in real civil engineering and geotechnical projects. The main advantage is the possibility of continuously measuring selected physical quantities over length to fulfil one of the fundamental goals for structural health monitoring: direct damage detection and estimation of its size. However, to utilise the advantages of DFOS approach, a set of requirements must be understood and implemented to obtain high-quality and reliable data.

The versatility of this technique lies in the ability to customise system elements (including the type of sensor, datalogger parameters, light scattering, installation methods, software and post-processing algorithms) according to the individual needs of a given project. In reference to reinforced concrete structures, it is essential to be aware that they operate in a cracked state and that a reliable assessment of local discontinuities (cracks) of a very small size (usually < 1 mm) is one of the primary challenges for the DFOS system.

The lessons learned while monitoring cracked concrete structures were summarized in the article to highlight what features of the DFOS tools are the most favourable from the crack analysis point of view. During the selection, not only their internal design should be considered, but also the strain range and resolution, axial stiffness, cross-section area, elasticity modulus, bending radius, material and its resistance to corrosion, durability, no need for pre-tensioning, the tendency to wave and more.

Theoretical considerations were supported by laboratory tests on full-size components. The novelty of the research lies in direct comparison between specific monolithic sensors and layered cables commercially available on the market, and often used for the same purposes. The obtained results have, thus, practical meaning, showing significant differences in the performance of different DFOS tools.

The findings include that layer-free sensors are adequate solution for crack analysis. The measuring fibre does not slip inside the cross-section, as it does in layered cables, providing that the structural strains are transferred in an as undisturbed way as possible. Based on the performed measurements, a reliable interpretation of the deformation state of the structure was possible.

The DFOS-based monitoring of cracked concrete structures is a challenging process. In addition to the monolithic sensors, which are one of the key components of the entire system, the appropriate high-spatial-resolution technique should be selected (preferably, Rayleigh-based), as well as installation methods which directly influence the physical interpretation of the obtained data. This article is a part of a discussion about the best DFOS configuration, which could be recommended for application in cracked concrete, depending on whether we are dealing with new or existing structures. A set of good practices based on lessons learned while applying monolithic DFOS strain sensors were presented in this article and discussed through laboratory and field proofs.

## Figures and Tables

**Figure 1 sensors-22-08713-f001:**
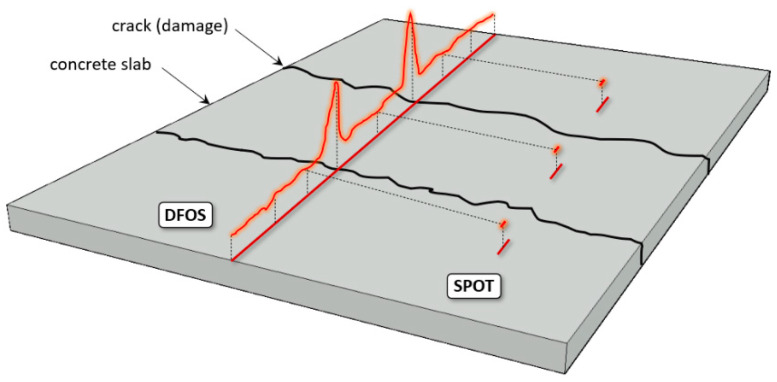
Comparison between capabilities of distributed fibre optic sensing DFOS and spot measurements in crack detection [8].

**Figure 2 sensors-22-08713-f002:**
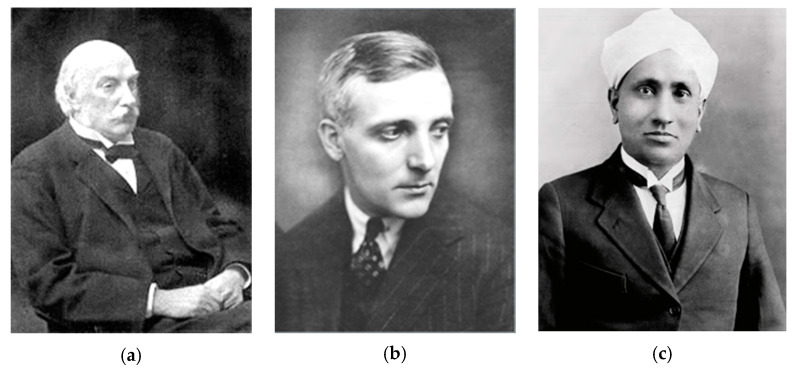
(**a**) John William Strutt (Lord Rayleigh); (**b**) Léon Brillouin; (**c**) Chandrasekhara Venkata Raman [6].

**Figure 3 sensors-22-08713-f003:**
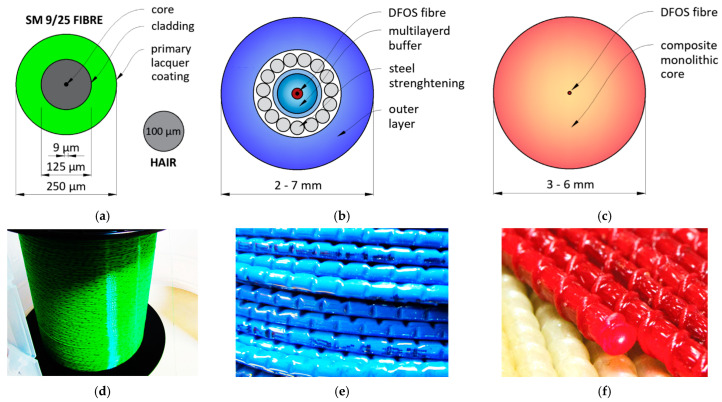
Example cross-sections of (**a**) single-mode optical fibre in its primary coating, (**b**) layered sensing cable with steel strengthening insert and (**c**) monolithic strain sensor with the corresponding views of their external surfaces (**d**–**f**).

**Figure 4 sensors-22-08713-f004:**
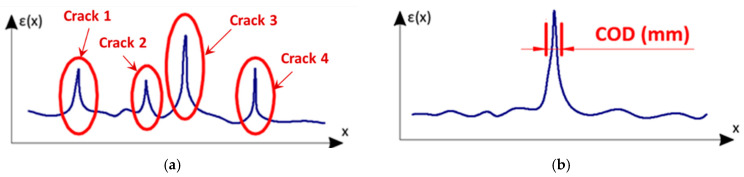
The scheme for crack analysis based on the DFOS system: (**a**) qualitative analysis (detection, location); (**b**) quantitative analysis (width estimation) [6].

**Figure 5 sensors-22-08713-f005:**
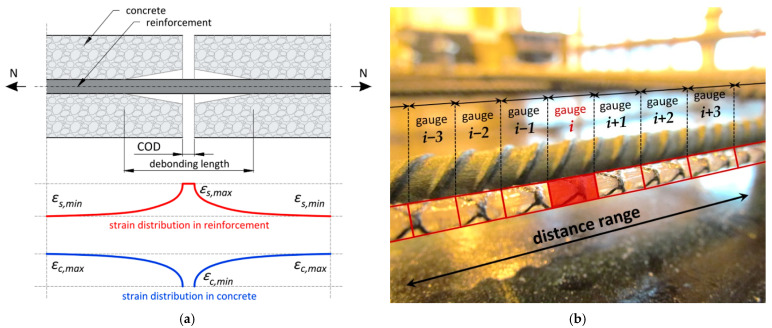
(**a**) Simplified scheme for a bond-slip crack model for reinforced concrete [51]; (**b**) simplified graphical interpretation of spatial resolution in distributed sensing.

**Figure 6 sensors-22-08713-f006:**
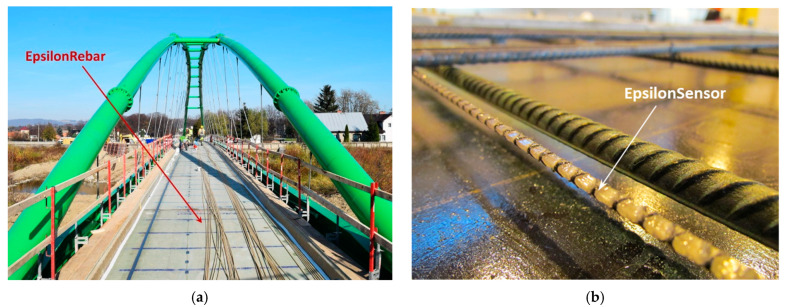
(**a**) Footbridge reinforced with stiff monolithic sensors; (**b**) concrete slab with the flexible monolithic sensor not influencing the structural behaviour [8].

**Figure 7 sensors-22-08713-f007:**
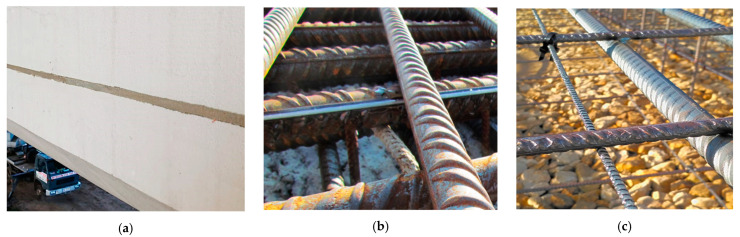
Possibilities of the sensors’ installation methods: (**a**) installation in near-to-surface grooves; (**b**) surface gluing (on steel bar); (**c**) embedding inside the concrete.

**Figure 8 sensors-22-08713-f008:**
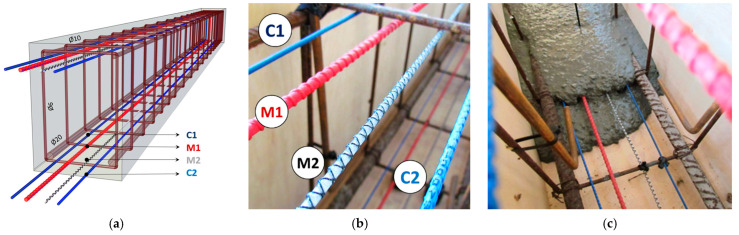
(**a**) Spatial visualisation of the reinforced concrete beam equipped with DFOS tools; (**b**) close-up to the selected tools; (**c**) the view of the beam during concreting.

**Figure 9 sensors-22-08713-f009:**
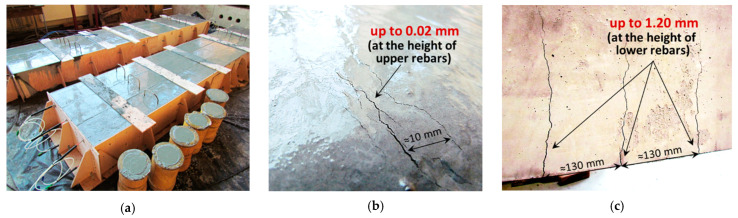
(**a**) The view of the beams after concreting; (**b**) thermal-shrinkage cracks within the upper part; (**c**) mechanical cracks within the lower part during bending tests.

**Figure 10 sensors-22-08713-f010:**
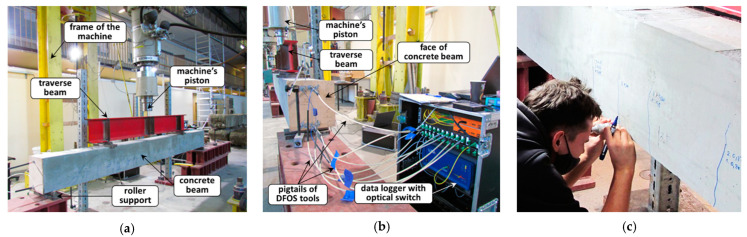
(**a**) Example beam during four-point bending test; (**b**) Rayleigh-based data logger with an optical switch connected to embedded DFOS tools; (**c**) reference readings.

**Figure 11 sensors-22-08713-f011:**
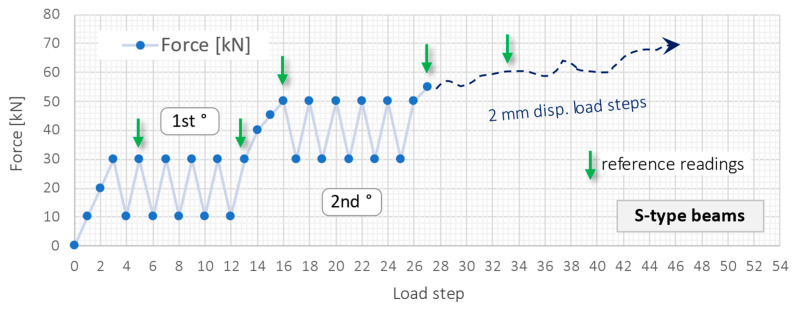
Designed research schedule for S-type beams.

**Figure 12 sensors-22-08713-f012:**
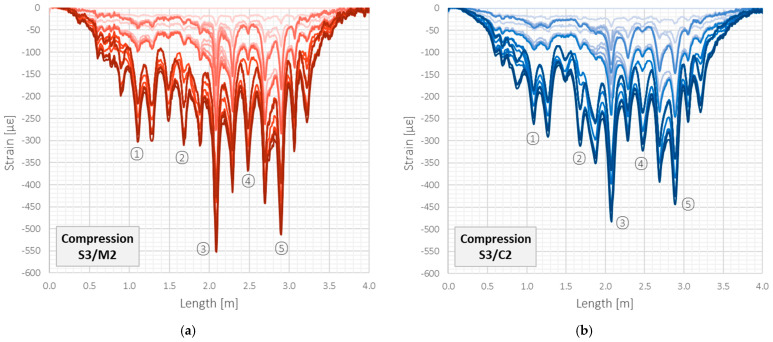
Beam S3/compression zone: strain profiles registered by (**a**) monolithic sensor M2 and (**b**) layered sensing cable C2 with steel strengthening insert.

**Figure 13 sensors-22-08713-f013:**
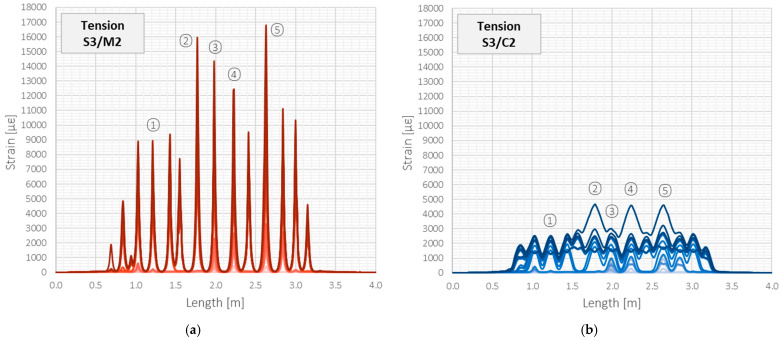
Beam S3/tension zone: strain profiles registered by (**a**) monolithic sensor M2 and (**b**) layered sensing cable C2 with steel strengthening insert.

**Figure 14 sensors-22-08713-f014:**
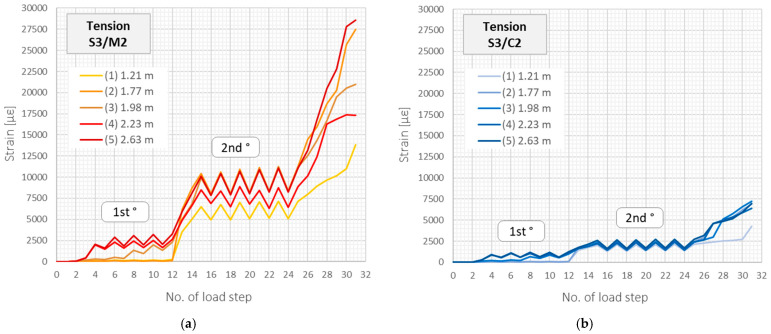
Beam S3/tension zone: crack-induced strain peaks over subsequent load steps registered by (**a**) monolithic sensor M2 and (**b**) layered sensing cable C2 with steel strengthening insert.

**Figure 15 sensors-22-08713-f015:**
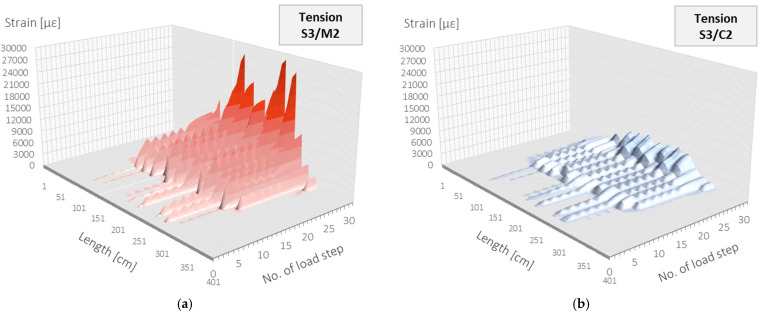
Beam S3/tension zone: strain profiles in length and load step domain registered by (**a**) monolithic sensor M2 and (**b**) layered sensing cable C2 with steel strengthening insert.

**Figure 16 sensors-22-08713-f016:**
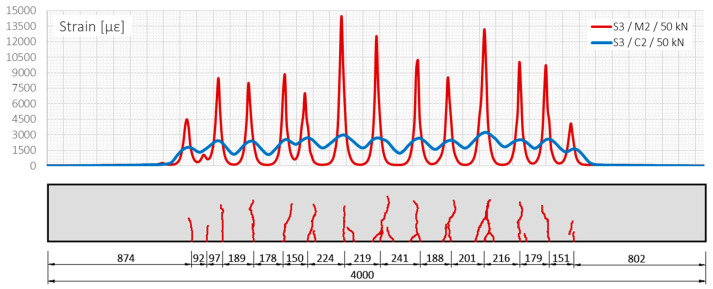
Strain profiles over length measured by monolithic sensor M2 and layered cable C2 (beam S3, load step no. 26, force = 50 kN).

**Figure 17 sensors-22-08713-f017:**
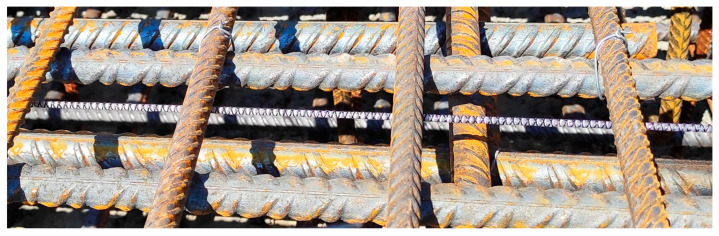
Installation of the monolithic sensor by tying to the existing reinforcement.

**Figure 18 sensors-22-08713-f018:**
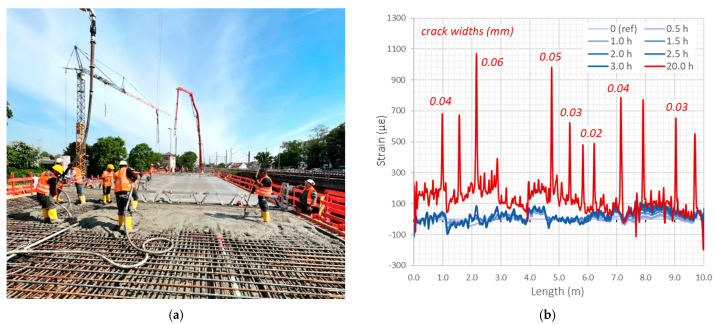
(**a**) The view of the slab during concreting; (**b**) example strain profiles with crack analysis 20 h after concreting.

**Figure 19 sensors-22-08713-f019:**
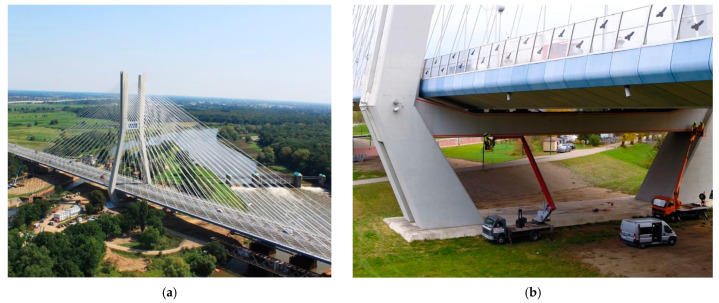
(**a**) Rędziński Bridge in Wrocław, Poland—general view (photo: W. Kluczewski); (**b**) the close-up of the lower crossbeam connecting two inclined legs of the pylon.

**Figure 20 sensors-22-08713-f020:**
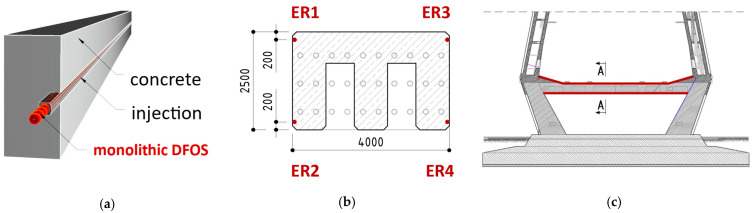
(**a**) Installation in near-to-surface grooves; (**b**) cross-section of the beam with locations of the sensors; (**c**) side view of the crossbeam with sensors marked over its length.

**Figure 21 sensors-22-08713-f021:**
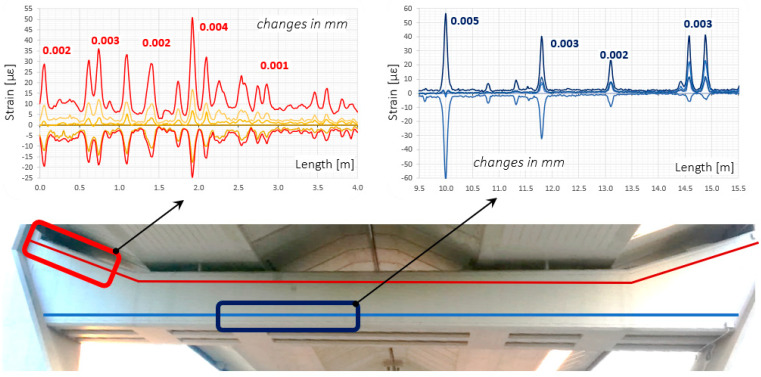
Example strain profiles over selected sections of the lower and upper monolithic sensor with crack width analysis.

**Figure 22 sensors-22-08713-f022:**
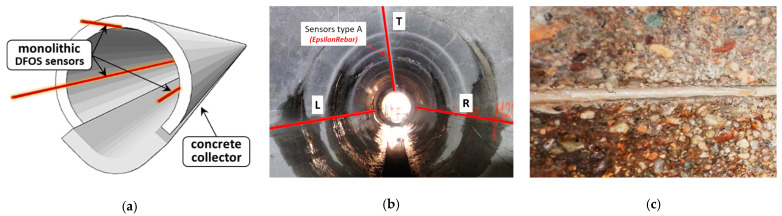
(**a**) Design of the collector with monolithic sensors; (**b**) the view of the collector with marked locations of monolithic sensors; (**c**) near-to-surface groove filled with injection.

**Figure 23 sensors-22-08713-f023:**
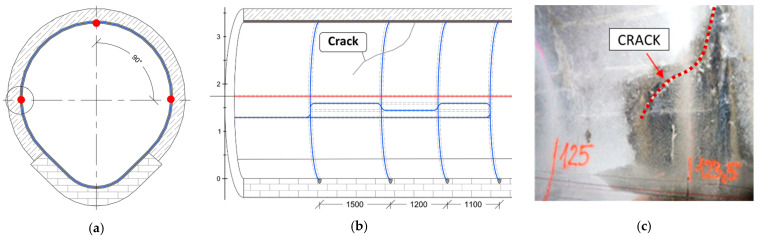
(**a**) Cross-section of the collector [58]; (**b**) location of the circumference monolithic sensor [58]; (**c**) the view of the crack documented before installation.

**Figure 24 sensors-22-08713-f024:**
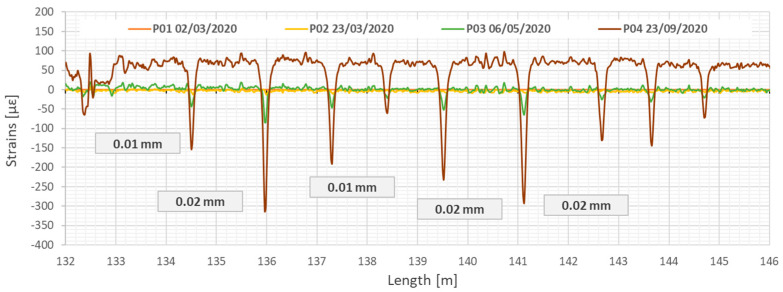
Strain profiles measured by the monolithic sensor (T) along the test section showing the primary discontinuities in the concrete collector closing during strengthening process. Example section from 132 to 146 m. Measurement P01—after installation of GRP panels, P02—during grout injection, P03—after completed grout injection, P04—after filling the collector with sewage [58].

**Table 1 sensors-22-08713-t001:** Comparison between the parameters of two different monolithic strain sensors [16].

Parameter	EpsilonSensor	EpsilonRebar
Strain resolution	±1 µε	±1 µε
Maximum strain	±40,000 µε (±4%)	±20,000 µε (±2%)
Standard diameter	Ø3 mm	Ø5 mm
Elastic modulus	3 GPa	50 GPa
Axial stiffness EA	21 kN	982 kN
Core material	PLFRP (polyester + epoxide)	GFRP (glass + epoxide)
Bending radius	50 mm	350 mm
Sensor weight	13 kg/km	45 kg/km
Light scattering ^1^	Rayleigh, Brillouin, Raman	
Delivery method	coils or straight sections	
Length	any length made to order	

^1^ Compatible with optical devices based on such phenomena.

**Table 2 sensors-22-08713-t002:** Design procedure of DFOS system for crack monitoring in concrete structures.

Step	Selection of	Details Considered
1	DFOS technique (Rayleigh, Brillouin, hybrid)	Measurement parameters (spatial resolution, accuracy, strain resolution, distance range, acquisition time, number of channels, etc.).
2	Monolithic strain sensor	Geometrical and mechanical properties (diameter, elastic modulus, strength, maximum strain, external braid, bending radius).
3	Installation method	Surface or near-to-surface installation for existing structures (with analysis of the adhesive’s properties). Embedding inside the new structures.
4	Thermal compensation	Raman technique, hybrid measurements, special DFOS temperature sensors, conventional spot temperature gauges (depending on distance range or expected temperature distributions and changes).
5	Post-processing algorithms	Data validation, thermal compensation algorithms, strain presentation, crack detection, width estimation, assessment of uncertainties, results visualisation (in length and time domain).

**Table 3 sensors-22-08713-t003:** Calculated crack widths based on strain profiles measured by monolithic sensor M2 and layered cable C2 (beam S3, load step no. 26, force = 50 kN).

Crack	Width (mm)			Diff. (mm)	Diff. (%)
	M2	C2	Ext. ref.	(M2 − C2)	(M2 − C2)/M2 × 100%
①	0.329	0.267	0.30	0.062	19.0
②	0.548	0.351	0.60	0.197	35.9
③	0.451	0.318	0.40	0.133	29.6
④	0.409	0.309	0.35	0.100	24.5
⑤	0.593	0.382	0.45	0.211	35.6
**mean**	**0.466**	**0.325**	**0.42**	**0.141**	**28.9**

## Data Availability

The data presented in this study are available on request from the corresponding author.

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
