# Peer review of "The Smart Nervous System for Cracked Concrete Structures: Theory, Design, Research, and Field Proof of Monolithic DFOS-Based Sensors"

_sensors, 2022, doi:10.3390/s22228713_

Round 1
Reviewer 1 Report
Comments:
This study presents research on the performance of composite and monolithic sensors for distributed fiber optic sensing. Laboratory tests and three fields test were conducted. The topic is interesting and the paper is well prepared in organization and written. However, there are some questions that need to be figured out to fully understand this paper, as shown following:
Line 57-59, it is not clear the reason why traditional SHM systems cannot answer the question about the state of the structure between the measurement points. Moreover, Fig 1 only shows one type of structural component: concrete slab. For other cases, the proposed methods may be not suitable, such as steel girder or concrete column. This is because the proposed sensor cannot be embedded in a steel structure. In addition, the damage to the concrete column is concentrated at the bottom or top area, other than the whole element. So please revise this paragraph with a more rigorous expression.
Line 102-103. The reviewer highly doubts the expression that the life of the proposed DFOS is comparable to the operation lifetime of the structure itself. Do you have any references to support your thoughts?
Line 112-114. The reviewer does not think it is necessary to give the pictures of three great people in the area of optical phenomena. As instead, the reviewer prefers to see the schematic diagram or principle of these methods.
Line 203. There is an interface between core optical fiber and outside composite. How to make sure the consistent deformation between them?
Fig 6 a. Could you give a figure with the epsilonrebar zoomed in? The authors highlighted four rebars, is that epsilonrebar sensor or something else? How many sensors did this project involve?
Line 452. What’s the meaning of S-type and L-type beam? Could you give the dimension info? From the fig 8 a, the cross-section of the beam should be rectangular, so it is confused.
Line 471. Please give the precision of the load cell, type or model and capacity. Because the author mentioned the testing machine with a force range up to 1000 kN. However, the applied load is pretty small, which is about 30 kN in the 1st phase.
Table 3. The authors have to give the measured width of the crack using the visual-based method to validate the effectiveness of the M2 sensors.
Line 595. How to convert the measured strain to crack width? It seems like the comparison results is not very good?
Line 462. Could the monolithic sensor measure the process of crack shrinkage? Could you give the results?
Author Response
Dear Reviewer, thank you for all your comments and suggestions to improve our paper. Please find attached our answers and explanations. We also provided appropriate corrections in the manuscript.
Kind regards,
Authors

Reviewer 2 Report
Attached file

Author Response

(The authors gave the same response as above.)

Reviewer 3 Report
The authors present performed a comparison study between monolithic sensors and layered cables embedded inside manufactured beams that were statically loaded under a four-point bending test. The results provide meaningful insights about distributed fiber optic sensing elements with applications of interest to the civil engineering community. In general, the paper is well-written and carefully organized. The reviewer has minor comments that may improve the quality of the manuscript provided below.
Figure 4. a: please label the circled peaks in red.
Figure 9b and c: provide a scale to accurately show the actual size of the cracks
Figure 10a and b: label the equipment and components shown in the photos
Figure 10c: provide a better-resolution photo
Figure 22a: kindly label the red rebars
Author Response

(The authors gave the same response as above.)

Round 2
Reviewer 2 Report
Congratulations for the corrections the article was very good